# Protective Effects of BNT162b2 Vaccination on Aerobic Capacity Following Mild to Moderate SARS-CoV-2 Infection: A Cross-Sectional Study Israel

**DOI:** 10.3390/jcm11154420

**Published:** 2022-07-29

**Authors:** Yair Blumberg, Michael Edelstein, Kamal Abu Jabal, Ron Golan, Neta Tuvia, Yuval Perets, Musa Saad, Tatyana Levinas, Dabbah Saleem, Zeev Israeli, Abu Raya Alaa, Gabby Elbaz Greener, Anat Amital, Majdi Halabi

**Affiliations:** 1Rivka Ziv Medical Center, Safed, Israel; michael.edelstein@biu.ac.il (M.E.); kamala@ziv.gov.il (K.A.J.); ron.g@ziv.gov.il (R.G.); netat@ziv.gov.il (N.T.); mozess1@hotmail.com (M.S.); tatyana.l@ziv.gov.il (T.L.); saleemdabbah@ziv.gov.il (D.S.); zeev.i@ziv.gov.il (Z.I.); alaa.abu@ziv.gov.il (A.R.A.); anatam@ziv.gov.il (A.A.); 2Azrieli Faculty of Medicine, Bar-Ilan University, Safed 1311502, Israel; yuval.p151@gmail.com (Y.P.); gabbyelbaz@yahoo.com (G.E.G.); 3Department of Cardiology, Hadassah Medical Center, Jerusalem 9112001, Israel

**Keywords:** SARS-CoV-2, COVID-19, aerobic capacity, vaccination

## Abstract

Patients previously infected with acute respiratory syndrome coronavirus 2 (SARS-CoV-2) may experience post-acute adverse health outcomes, known as long COVID. The most reported symptoms are fatigue, headache and attention/concentration issues, dyspnea and myalgia. In addition, reduced aerobic capacity has been demonstrated in both mild and moderate COVID-19 patients. It is unknown whether COVID-19 vaccination mitigates against reduced aerobic capacity. Our aim was to compare the aerobic capacity of vaccinated and unvaccinated individuals previously infected with SARS-CoV-2. Methods: Individuals aged 18 to 65 years with laboratory-confirmed mild to moderate COVID-19 disease were invited to Ziv Medical Centre, Israel, three months after SARS-CoV-2 infection. We compared individuals unvaccinated at the time of infection to those vaccinated in terms of aerobic capacity, measured using symptom-limited cardiopulmonary exercise test (CPET). Results: We recruited 28 unvaccinated and 22 vaccinated patients. There were no differences in baseline demographic and pulmonary function testing (PFT) parameters. Compared with unvaccinated individuals, those vaccinated had higher V’O_2_/kg at peak exercise and at the anaerobic threshold. The V’O_2_/kg peak in the unvaccinated group was 83% of predicted vs. 100% in the vaccinated (*p* < 0.002). At the anaerobic threshold (AT), vaccinated individuals had a higher V’O_2_/kg than those unvaccinated. Conclusions: Vaccinated individuals had significantly better exercise performance. Compared with vaccinated individuals, a higher proportion of those unvaccinated performed substantially worse than expected on CPET. These results suggest that vaccination at the time of infection is associated with better aerobic capacity following SARS-CoV-2 infection.

## 1. Introduction

People who were infected with severe acute respiratory syndrome coronavirus 2 (SARS-CoV-2) may experience post-acute adverse health outcomes. When symptoms persists more than three months post-acute infection and last more than two months, the phenomenon is named long COVID-19 (long COVID) [1]. Symptoms of long COVID may fluctuate or relapse over time and can affect activities of daily living. At least 30% of patients infected with SARS-CoV-2 suffer from long COVID [1]. The most commonly reported symptoms are fatigue, headache and attention disorders [2]. In addition, reduced aerobic capacity has been demonstrated in both mild and moderate COVID-19 patients [3]. The pathophysiological mechanisms are still poorly understood and may include direct viral toxicity, inflammation, autoimmune response and thrombosis vasculitis, which might also lead to exercise intolerance [4]. The impact of vaccination on long COVID symptoms remains unclear. Current evidence suggest that vaccination may play a protective role against at least some long COVID symptoms: An evidence review undertaken by the UK Health Security Agency suggests that infected individuals vaccinated against COVID-19 are less likely than those unvaccinated to report long COVID. The magnitude of the protective effect, however, remains unclear [5]. Most available studies measured self-reported symptoms [6]. In previous studies, it was shown that CPET is an adequate clinical tool for estimating long COVID severity [7]. There is little evidence available regarding whether there is an association between COVID-19 vaccination and aerobic capacity post SARS-CoV-2 infection. The aim of our study was to compare aerobic capacity and exercise performance in individuals who were vaccinated at the time of their SARS-CoV-2 infection to those who were infected but unvaccinated at the time.

## 2. Methods

This prospective cross-sectional study was conducted at the cardiac rehabilitation department of Ziv Medical Centre, a 300-bed government hospital in Safed, Northern Israel, between March 2021 and April 2022. Individuals aged 18 to 65 years with laboratory-confirmed mild to moderate COVID-19 disease according to the National Institutes of Health definitions and no severe pre-existing cardiac or respiratory condition were eligible for participation [8]. The study was advertised through hospital clinics and among staff and students based at the hospital. In order to detect a difference of 25 L/min in minute ventilation (VE) with a power of 0.8 and an alpha of 0.05, 17 patients were required in each of the two groups.

Information about demographics and prior health issues was collected among all participants prior to the exercise test.

## 3. Cardiopulmonary Exercise Test

Each participant performed a symptom-limited cardiopulmonary exercise test (CPET) using an individually calibrated bicycle ergometer protocol, according to Albouaini et al. [9]. Prior to exercise, each patient underwent Spirometry tests according to the American Thoracic Society, and at least three acceptable measurements were obtained per participant [10].

The CPET test was performed on a cycle ergometer (Cortex-Medical), and subjects were asked to maintain a constant pedalling frequency of 60 ± 5 revolutions/minute. Throughout the test, cardiac electrical activity was monitored using continuous electrocardiography, and blood pressure and Perceived Exertion (RPE) were measured every two minutes.

The peak oxygen consumption (V’O_2_) is defined as the highest value of V’O_2_ attained in a 20 s interval. The anaerobic threshold (AT), referring to the point at which ventilation starts to increase at a faster rate than V’O_2_, was determined by the V-slope method [9]. The minute ventilation/carbon dioxide production (V’E/V’CO_2_) slope is calculated as the coefficient of linear regression obtained by plotting the V’E and V’CO_2_ data of the subject’s exercise phase. The oxygen uptake efficiency slope (OUES) was calculated according to Baba et al.’s method [11]. For each individual, we compared the observed values to predicted values based on age, sex and height, which are calculated by the Hansen and Wasserman predicted value of exercise testing [12].

## 4. Statistical Analysis

We used the Shapiro–Wilk test to check that the distribution of the main outcome (VO2) was normal (*p* = 0.11). We therefore used parametric tests for our analysis. All results are presented as mean ± standard error (SE). The vaccinated and unvaccinated groups were also compared in terms of difference in CPET values and in terms of difference in the % of predicted values, using t-tests for comparisons between unvaccinated and vaccinated CPET parameters. A *p*-value of 5% or less was considered statistically significant.

The analysis was performed using SPSS software (IBM SPSS Statistics for Windows, Version 25.0., Armonk, NY, USA). The study protocol was approved by the Ziv Hospital’s Ethics Committee; ethics approval: 0100-20-ZIV.

## 5. Results

Fifty participants were enrolled: 28 were unvaccinated at the time of infection (referred to as unvaccinated) and 22 had breakthrough infection, i.e., vaccinated at the time of infection (referred to as vaccinated). All vaccinated patients had received at least two doses, with the exception of two patients. Both groups had comparable gender distributions and mean height, weight and body mass index (*p* > 0.8 for all, Table 1), but vaccinated patients were younger (mean age, 41 ± 9 vs. 47 ± 12 years). Spirometry measurements at baseline were comparable in the unvaccinated and vaccinated groups, as presented in Table 1. The smoking habits are comparable in both groups, as presented in Table 1.

The CPET test was conducted at a mean of 119 ± 24 days after acute diseases with no difference in follow-up time between the groups (*p* = 0.12). The major peak cardiopulmonary metrics are summarized in Table 2.

Compared with unvaccinated individuals, those who were vaccinated had higher mean V’O2, V’O_2_/kg and heart rate (HR) at peak (Figure 1B, Table 2). The mean peak V’O_2_/kg in the unvaccinated group was 83% of the predicted group vs. 100% in the vaccinated group (*p* < 0.002, Table 2). In the unvaccinated group, 14/28 subjects (50%) had a V’O_2_ peak <80% of predicted vs. 2/22 (9%) among the vaccinated group (Figure 2A). The maximum HR was reduced among unvaccinated participants compared with those who were vaccinated (Figure 1B, Table 2).

The VE at peak was significantly different between the groups, with a percentage of predicted values of 74.7 ± 19.5 in the unvaccinated group and 93 ± 18 in the vaccinated group (*p* < 0.001, Table 2). Twenty of twenty-eight unvaccinated participants (70%) had <80% of predicted of VE, compared with only four of twenty-two (18%) in the vaccinated group (Figure 2B). There was a significant difference between groups in VE/VO_2_ at peak exercise and no significant difference between the two groups in the OUES (Table 2).

There was a significant difference in work rate at peak exercise reaching 112 ± 41.7 Watts in the vaccinated group compared to 130 ± 40 in the unvaccinated group (*p* < 0.04, Table 2)

Table 3 summarizes the cardiopulmonary at the anaerobic threshold (AT).

There was a significant difference in the subjective feeling (RPE), which was observed at the AT of 13.7 ± 3.2 in the unvaccinated group versus 11.3 ± 3 in the vaccinated group, while at peak exercise, both groups reported a feeling of very high effort (almost maximal) (RPE 19.2 ± 1 versus 18.8 ± 2).

When comparing the AT (Table 3), there was a significant difference between the groups in the absolute values (VO_2_/kg) that are reached at this point. Furthermore, the percentage of peak VO_2_/Kg was higher for the unvaccinated group versus vaccinated group (71 ± 12 versus 63 ± 8).

## 6. Discussion

Our study showed a difference in aerobic capacity between individuals who were vaccinated and unvaccinated at the time of infection with SARS-CoV-2 three months after recovering from the acute phase of COVID-19. The work rate was significantly different between vaccinated and unvaccinated individuals at the peak of exercise and at the AT, indicating a lower aerobic efficiency.

In our study, we observed a lower maximum HR in the unvaccinated group, suggesting chronotropic incompetence that was previously described as contributing to limited exercise capacity [13,14].

On average, the unvaccinated group reached a lower proportion of their predicted peak V’O_2_; 50% reached less than 80% of the predicted VO_2_ compared with only 9% in the vaccinated group.

Seventy percent of the unvaccinated individuals reached less than 80% of predicted VE. In addition, both groups had a normal VE/VCO_2_ slope, which reflects the increase in ventilation in response to CO_2_ production and, thus, shows increased ventilatory drive. It was previously shown that, in some studies, hyperventilation could be an explanation for long COVID [15]; however, in our study, we observed hypoventilation, with vaccinated participants reaching almost 96% of predicted compared with 73% in the unvaccinated participants. This is in accordance with studies that show normal or low ventilation in long COVID [16]. In our study, there is a significant difference in VE/VO_2_ at the peak of exercise (Table 1). This could explain the inability to reach the expected VE in the unvaccinated group due to insufficient O_2_ delivery to the working muscle, which could be because of chronotropic incompetence and autonomic nerve system that leads to a lower heart rate and ventilation.

In our cohort, we show that there is also a difference in ventilation (V’E), with lower values for the unvaccinated group, as was shown in other pathologies such as heart failure and metabolic diseases [17,18].

The AT was significantly different between the vaccinated and unvaccinated groups, with the AT appearing later for vaccinated participants. The AT is the point at which there is a switch from aerobic to anaerobic metabolism, which results in differences in the ventilation pattern [19]. This difference was observed in our results. Interestingly, there was also a difference between the groups in terms of their subjective feeling at this point, which warrants further research.

Identifying the pathological mechanism leading to an inability to increase HR and ventilation is beyond the scope of this study. Suggested mechanisms in the literature include immune-mediated damage to the autonomic nervous system during COVID-19 and a peripheral cardiac limitation to exercise resulting from an oxygen-diffusion defect [20,21].

Our study has some limitations, including the small cohort size and difference in age at baseline. However, the fact that we compared findings in individuals with their own predicted value adjusts for this difference to a large extent. Two of the patients were only partially vaccinated, possibly leading to a slight underestimation of the effect of vaccination. We did not evaluate blood gas, which could have revealed more about the main cause of exercise limitations for patients with reduced pVO_2_.

## 7. Conclusions

This study suggests that COVID-19 patients can suffer from objective limitations to exercise capacity in the months following their acute episode. Our study is the first to show a protective effect of vaccinations against decreased aerobic capacity. As a more objectively quantifiable definition of long COVID is needed, studies able to demonstrate measurable changes post-acute infection with SARS-CoV-2 are essential. The measured protective effect of vaccination provides additional reasons to continue to intensify the vaccine drive globally. Similar studies should be replicated on a larger scale to confirm our results.

## Figures and Tables

**Figure 1 jcm-11-04420-f001:**
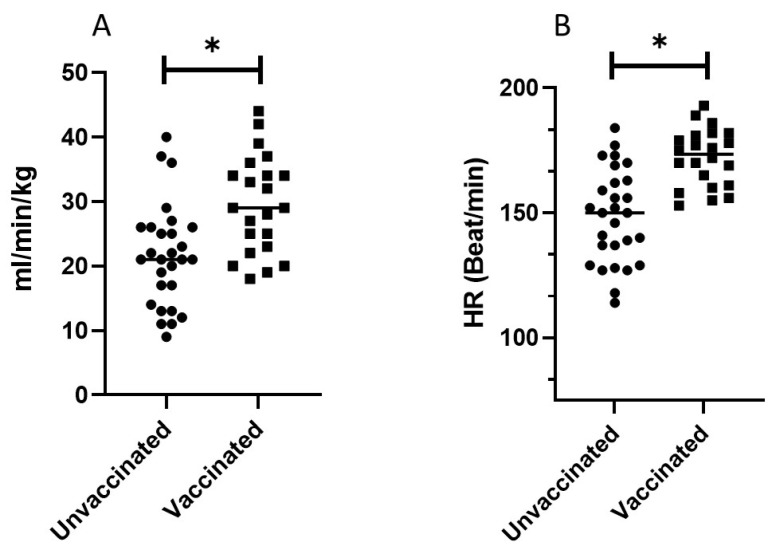
VO_2_/kg peak (**A**) and peak heart rate (**B**) among COVID-19 unvaccinated vs. vaccinated patients previously infected with SARS-CoV-2 (* *p* < 0.05).

**Figure 2 jcm-11-04420-f002:**
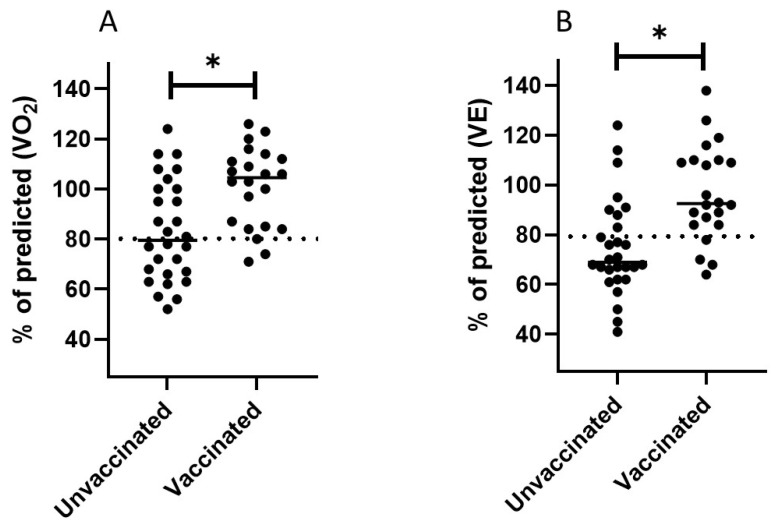
Percentage of predicted value peak V’O_2_ (**A**) and ventilation (VE) (**B**). The dashed line represents 80% of the predicted value (* *p* < 0.05).

**Table 1 jcm-11-04420-t001:** Participant characteristics, symptoms and pulmonary function test (PFT).

	Unvaccinated (*n* = 28)	Vaccinated (*n* = 22)
Male (*n*, %)	15 (53%)	10 (45%)
Female (*n*, %)	13 (46%)	12 (55%)
Age (years, SE)	47 ± 12	41 ± 9
Height, (cm, SE)	170 ± 8	169 ± 11
Weight, (kg, SE)	82.5 ± 16	76 ± 12
Body Surface Area	1.8 ± 0.6	1.9 ± 0.22
Body mass Index	28 ± 6	26 ± 3.3
Comorbidities		
Diabetes Mellitus (*n*, %)	4 (14%)	-
Hypertension (*n*, %)	4 (14%)	-
Smoker (*n*, %)	3 (11%)	4 (18%)
PFT (Pulmonary Function Test)		
Forced Vital Capacity (FVC)	4.5 ± 1.2	4.8 ± 1.5
Forced Expiratory Volume (FEV_1_)	3.5 ± 0.9	3.7 ± 1.1
FEV_1_/FVC	79 ± 8	79.5 ± 7.2

SE—Standard error, FVC—Forced vital capacity, FEV1—Forced expiratory volume in 1 s.

**Table 2 jcm-11-04420-t002:** CPET parameters at peak exercise.

	Unvaccinated		Vaccinated		*p* Value	
	Measured	% of Predicted	Measured	% of Predicted	Difference in Measured Values Unvaccinated vs. Vaccinated	Difference in % Predicted Values Unvaccinated vs. Vaccinated
**V’O_2_ (L/min)**	1.8 ± 0.7	83.2 ± 20	2.22 ± 0.9	100 ± 16	0.04	0.002
**V’O_2_/kg (mL/min/kg)**	21.6 ± 8	83.2 ± 20	27.8 ± 8	100 ± 16	0.04	0.002
**V’O_2_/HR (mL)**	11.8 ± 3.8	85 ± 15	12.7 ± 4	93 ± 15	0.39	0.051
**WR (Watt)**	112 ± 41.7	63.8 ± 21	130 ± 40	74 ± 19	0.04	0.06
**HR (bit/min)**	148.5 ± 18.8	96 ± 11	174 ± 12.5	108 ± 8	0.01	0.01
**VE (L/min)**	70.3 ± 26	74.7 ± 19.5	97 ± 33	93 ± 18	0.001	0.001
**BF (Vt/min)**	40.1 ± 7.3	26 ± 26.71	50.5 ± 7.5	149 ± 25	0.002	0.003
**OUES**	2.1 ± 0.6	73 ± 17	2.3 ± 0.9	83 ± 26	0.31	0.12
**BR (L/min)**	47.1 ± 20.6	-	31.29 ± 17	-	0.002	-
**RPE**	18.8 ± 2	-	19.2 ± 1	-	0.37	-
**VE/VO_2_**	37.1 ± 6.5	-	41.5 ± 4.5	-	0.01	-
**VE/VCO_2_**	35.4 ± 6	-	37.8 ± 4	-	0.17	-
**VE/VCO_2_ slope**	36.5 ± 7	-	34 ± 4	-	0.14	-

V’O_2_—Oxygen consumption, V’O_2_/kg—Oxygen consumption minute per kilogram of body weight, V’O_2_/HR—Oxygen-pulse, WR—Work rate, HR-Heart Rate, VE—Minute ventilation, OUES—Oxygen uptake efficiency slope, BF—Breathing frequency, BR—Breathing reserve, RPE—Rating of Perceived Exertion.

**Table 3 jcm-11-04420-t003:** Measurements at anaerobic threshold (AT).

	Unvaccinated		Vaccinated		*p* Value	
	Measured	% of Peak	Measured	% of Peak	Difference in Measured Values Unvaccinated vs. Vaccinated	Difference in % Peak Unvaccinated vs. Vaccinated
**V’O_2_ (L/min)**	1.1 ± 0.5	71 ± 12	1.4 ± 0.5	63 ± 8	0.07	0.02
**V’O_2_/kg (mL/min/kg)**	14.2 ± 3	71 ± 12	18.4 ± 5.5	63 ± 8	0.001	0.02
**V’O_2_/HR (mL)**	9.3 ± 2	83 ± 9	11 ± 4	85 ± 9	0.12	0.39
**WR (Watt)**	60 ± 23	61.2 ± 15	67 ± 13	52 ± 10	0.2	0.01
**HR (bit/min)**	120 ± 13.5	75 ± 12	131 ± 20	75 ± 6	0.05	0.8
**VE (L/min)**	36.5 ± 8	61.4	43 ± 18	43 ± 5	0.18	0.01
**BF (Vt/min)**	29.2 ± 6	78.7 ± 26	27 ± 7	58 ± 18	0.43	0.03
**BR (L/min)**	73 ± 20	-	84 ± 24	-	0.1	-
**RPE**	13.7 ± 3.2	-	11.3 ± 3	-	0.001	-
**VE/VO_2_**	31 ± 9	83 ± 13	28 ± 6	67 ± 8	0.12	0.003
**VE/VCO_2_**	32 ± 9	88 ± 10	28 ± 2.5	74 ± 8	0.07	0.001

V’O_2_—Oxygen consumption, V’O_2_/kg—Oxygen consumption minute per kilogram of body weight, V’O_2_/HR—Oxygen-pulse, WR—Work rate, HR—Heart Rate, VE—Minute ventilation, BF—Breathing frequency, BR—Breathing reserve, RPE—Rating of Perceived Exertion.

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
