# Peer review of "Protective Effects of BNT162b2 Vaccination on Aerobic Capacity Following Mild to Moderate SARS-CoV-2 Infection: A Cross-Sectional Study Israel"

_jcm, 2022, doi:10.3390/jcm11154420_

Round 1

Reviewer 1 Report

1. Are continuous outcomes in this manuscript were passed the normality test (normal distribution)?

If it's not a normal distribution. Suggest using a non-parametric statistic for continuous (Mann-Whitney test) with median (IQR) instead of a parametric statistic (t-test with a mean (±SD)).

2. How to make sure that the participants in this study got the first SAR-CoV-2 infection, not a second infection.

3. According to Diabetes 2022;71(Supplement_1):174-LB (https://diabetesjournals.org/diabetes/article/71/Supplement_1/174-LB/146042). It seems diabetes mellitus could increase the risk of long COVID-19. However, it is controversial about the risk factors that increased long COVID-19.

Please check the outcomes of diabetes mellitus (also hypertension) in individuals of the unvaccinated group. If it "outlier", you can exclude these points or divide them into another group in the supplementary.

Author Response

Response to Reviewer 1 Comments

  1. Are continuous outcomes in this manuscript were passed the normality test (normal distribution)? If it's not a normal distribution. Suggest using a non-parametric statistic for continuous (Mann-Whitney U test) with median (IQR) instead of a parametric statistic (t-test with a mean (±SD))

Response 1: We checked the normality of the distribution of the main continuous outcome (VO2) using a Shapiro-Wilk test which shoed the          distribution was normal (p=0.11)

  1. How to make sure that the participants in this study got the first SAR-CoV-2 infection, not a second infection.

Response 2: All participants were asked to present a certificate of infection for           COVID-19, and we were able to check that this was their first infection, based    on their electronic health records.

  1. According to Diabetes 2022;71(Supplement_1):174-LB (https://diabetesjournals.org/diabetes/article/71/Supplement_1/174-LB/146042). It seems diabetes mellitus could increase the risk of long COVID-19. However, it is controversial about the risk factors that increased long COVID-19. Please check the outcomes of diabetes mellitus (also hypertension) in individuals of the unvaccinated group. If it "outlier", you can exclude these points or divide them into another group in the supplementary

Response 3: We rechecked the outcomes for the diabetic and hypertension     patients- none were outliers.All the participants with diabetes or hypertension        received appropriate pharmaceutical treatment and their conditions were well        controlled. Therefore, we don't think that these participants should be   excluded from the results.

Reviewer 2 Report

1-      It is not correct to include study period in scientific article title.

2-      This word ( Exercise) is not keyword.

3-      Long abstract, reduce it.

4-      at the time vs at same period or within same period.

5-      (We found …) in article form it is not correct to use we and it is not correct start for a section.

6-       In article form there is no Limitations as a section. It can be included within discussion text.

Author Response

Response to Reviewer 2 Comments

  1. It is not correct to include study period in scientific article title.

Response 1: The article title was changed

  1. This word (Exercise) is not keyword.

Response 2:Thank you, the word was deleted from the list of keywords.

  1. Long abstract, reduce it

Response 3: Thank you very much, we shortened the abstract.

  1. at the time vs at same period or within same period.

Response 4: We checked the use of these terms in the text, and they were        appropriate. At the time of infection refers to their vaccination status when    infection occurs.

  1. (We found …) in article form it is not correct to use we and it is not correct start for a section

Response 5: Thank you very much; the sentence was changed to “Our study             showed “

  1. In article form there is no Limitations as a section. It can be included within discussion text

Response 6: Thank you for this comment. We merged the limitations section as part of the             discussion

Reviewer 3 Report

I congratulate the authors, the work is well structured, scientifically valid and the conclusions very interesting. The study is relevant given that it shows how SARS-CoV-2 infection in patients with long-covid reduces performance to physical exertion and the vaccine can be protective. I have only a few minor considerations.

1)      In the description of the data collected between the two cohorts of patients I see no reference to smoking habits. Were all the patients non-smokers? or was there some smoker? Perhaps a brief description should be included just to clarify if none of the subjects studied were smokers.

2)      2) Always with reference to the characteristics of the two cohorts of patients, a very brief description should be included regarding the possible condition of patients with regard to asthma. I imagine that none of the patients suffered, or ever suffered from asthma, it would be useful in my opinion to specify it.

Author Response

Response to Reviewer 3 Comments

  1. In the description of the data collected between the two cohorts of patients I see no reference to smoking habits. Were all the patients non-smokers? or was there some smoker? Perhaps a brief description should be included just to clarify if none of the subjects studied were smokers.

Response 1: We add the smoking habits to table 1

  1. Always with reference to the characteristics of the two cohorts of patients, a very brief description should be included regarding the possible condition of patients with regard to asthma. I imagine that none of the patients suffered, or ever suffered from asthma, it would be useful in my opinion to specify it.

Response 2: None of the participants had asthma as it can be seen from the PFT measurements that are presented in table 1.
